# The Liver Microbiome Is Implicated in Cancer Prognosis and Modulated by Alcohol and Hepatitis B

**DOI:** 10.3390/cancers12061642

**Published:** 2020-06-21

**Authors:** Jaideep Chakladar, Lindsay M. Wong, Selena Z. Kuo, Wei Tse Li, Michael Andrew Yu, Eric Y. Chang, Xiao Qi Wang, Weg M. Ongkeko

**Affiliations:** 1Division of Otolaryngology-Head and Neck Surgery, Department of Surgery, University of California San Diego, La Jolla, CA 92093, USA; jchaklad@ucsd.edu (J.C.); lmw010@ucsd.edu (L.M.W.); wtl008@ucsd.edu (W.T.L.); 2Research Service, VA San Diego Healthcare System, San Diego, CA 92161, USA; 3Department of Medicine, Columbia University Medical Center, New York, NY 10032, USA; szk9009@nyp.org; 4Department of Internal Medicine, Emory University School of Medicine, Atlanta, GA 30322, USA; michael.yu@emory.edu; 5Department of Radiology, University of California, San Diego, CA 92093, USA; e8chang@health.ucsd.edu; 6Radiology Service, VA San Diego Healthcare System, San Diego, CA 92161, USA; 7Department of Biomedical Sciences, College of Medicine, Florida State University, Tallahassee, FL 32306, USA; XiaoQi.Wang@med.fsu.edu

**Keywords:** hepatocellular carcinoma, microbiome, hepatitis B, alcohol, TCGA

## Abstract

Hepatocellular carcinoma (HCC) is one of the deadliest cancers in the world. Previous studies have identified the importance of alcohol and hepatitis B (HBV) infection on HCC carcinogenesis, indicating synergy in the methods by which these etiologies advance cancer. However, the specific molecular mechanism behind alcohol and HBV-mediated carcinogenesis remains unknown. Because the microbiome is emerging as a potentially important regulator of cancer development, this study aims to classify the effects of HBV and alcohol on the intratumoral liver microbiome. RNA-sequencing data from The Cancer Genome Atlas (TCGA) were used to infer microbial abundance. This abundance was then correlated to clinical variables and to cancer and immune-associated gene expression, in order to determine how microbial abundance may contribute to differing cancer progression between etiologies. We discovered that the liver microbiome is likely oncogenic after exposure to alcohol or HBV, although these etiological factors could decrease the abundance of a few oncogenic microbes, which would lead to a tumor suppressive effect. In HBV-induced tumors, this tumor suppressive effect was inferred based on the downregulation of microbes that induce cancer and stem cell pathways. Alcohol-induced tumors were observed to have distinct microbial profiles from HBV-induced tumors, and different microbes are clinically relevant in each cohort, suggesting that the effects of the liver microbiome may be different in response to different etiological factors. Collectively, our data suggest that HBV and alcohol operate within a normally oncogenic microbiome to promote tumor development, but are also able to downregulate certain oncogenic microbes. Insight into why these microbes are downregulated following exposure to HBV or alcohol, and why the majority of oncogenic microbes are not downregulated, may be critical for understanding whether a pro-tumor liver microbiome could be suppressed or reversed to limit cancer progression.

## 1. Introduction

Hepatocellular carcinoma (HCC) is the most common form of liver cancer and the second deadliest cancer, affecting over 500,000 people worldwide every year [1]. One well-established risk factor for HCC is chronic hepatitis B (HBV) infection, which accounts for approximately 50% of all cases [2]. In the absence of diagnostic markers for the early detection of the disease, HBV-related HCC exhibits extremely poor prognosis, with a median survival of less than 16 months [3]. Another well-established risk factor is excessive alcohol use. Studies have shown that alcohol consumption significantly increases the risk of HCC in HBV-related patients [4]. Multiple epidemiologic and pathologic studies have reported the synergism between alcohol and HBV infection in the context of HCC [5,6,7]. Despite the recent advancements in the knowledge of HCC and cancer in general, the molecular effects of ethanol exposure on the microenvironment and on HCC pathogenesis and progression, and its specific effects in HCC with a chronic viral hepatitis B background, remain poorly characterized.

The microbiome is defined as the genetic material of all the microbes—bacteria, fungi, protozoa and viruses—that live on and inside the human body. Since the microbiome is located primarily in the gut, it is not surprising that the gut microbiome has been the most studied and has been implicated in a wide variety of human diseases, including Alzheimer’s disease, cardiovascular disease, diabetes, arthritis, and cancer [8]. In addition, there is an overwhelming amount of evidence on how the gut microbiome regulates liver function, and consequently, liver diseases, including nonalcoholic fatty liver disease (NAFLD), cirrhosis, and hepatocellular carcinoma [9,10]. Since the liver receives 70% of its blood supply from the intestine, it acts as the first line of defense against gut-derived foreign entities. The gut bacteria play a critical role in the maintenance of gut-liver axis health, and breakdown in the homeostasis between bacteria and the host can lead to a disruption of the intestinal barrier function and may allow the migration of the bacteria from the intestinal lumen to other extraintestinal organs or sites. In fact, the derangement of the gut flora occurs in 20–75% of patients with chronic liver disease [10]. In contrast to our understanding of the gut microbiome, there have been very limited reports on the intratumor liver microbiome, especially in HCC. To date, there have been no etiology-specific studies focusing on the role of the liver microbiome in HCC pathogenesis and progression. Recent studies have shown that the microbiome is able to modulate the immune system and thereby alter homeostasis [11]. The altered abundance of specific microbes may increase or decrease immune activity [12]. Additionally, microbial metabolites may alter cell signaling pathways, thereby affecting cancer cells as well [13]. Through these mechanisms, microbes within the intratumoral liver microbiome may affect carcinogenesis.

In this study, we sought to expand the current understanding of the interplay among alcohol consumption, HBV, and HCC, using next-generation RNA-sequencing data from HBV+ HCC patients and adjacent normal liver tissues. We identified the most abundant bacterial microbiome based on etiology and divided our subjects into four cohorts: HBV− nondrinkers, HBV+ nondrinkers, HBV− drinkers, and HBV+ drinkers. We investigated the association between microbial diversity and abundance and their correlation with clinical variables, including survival, tumor stage, histologic grade, fibrosis Ishak score, and *Pugh classification* grade. We assessed whether alcohol consumption synergizes with HBV to regulate the microbiome and how intratumor HCC microbial dysbiosis in turn correlates with the clinical variables. In addition, we studied the correlation between microbial abundance and transcriptomic alterations. The results of this study suggest that both heavy alcohol use and HBV infection may utilize the intratumor microbiome to promote cancer development and that the microbiome likely plays previously uncharacterized roles in the HBV and alcohol-associated pathogenesis and progression of HCC.

## 2. Results

### 2.1. Isolation of Microbial Read Counts from Sequencing Data

Raw RNA-sequencing data were downloaded from The Cancer Genome Atlas (TCGA) database for liver hepatocellular carcinoma (LIHC) tumor samples and adjacent normal samples. The samples are divided into four cohorts: HBV+/drinker, HBV+/nondrinker, HBV−/drinker, and HBV−/nondrinker (Figure 1A). The Pathoscope direct alignment framework was applied to the RNA-sequencing data to determine the abundance of specific species, subspecies, or strains of microbes in each patient sample. Pathoscope first identifies any reads in the transcriptome that corresponds to human genomic sequences. Reads that do not map to any human sequences are then mapped to the genomes of bacteria. The NCBI Nucleotides database was used as the reference database for human and bacterial genome sequences.

### 2.2. Calculation of Alpha and Beta Diversities and Their Association with HBV or Alcohol Exposure

Alpha and beta diversities were calculated using the Qiime2 software suite [14]. We observed that, for several alpha diversity measures, including double observations, Fisher’s index, Gini index, and Brillouin’s index, we observed a significant difference between HBV+ and HBV− patients (Kruskal–Wallis, *p* < 0.05) (Figure 1B). However, no significant difference was observed between drinkers and nondrinkers (Figure 1C). For beta diversity, we observed that, for both etiologies, some clustering was observed between patients with the same etiology (Figure 1D,E). However, neither of the etiologies account for the full spectrum of beta diversity in LIHC.

### 2.3. Removal of Potential Contaminants

Because TCGA sample collection does not include a robust protocol for the sterile collection of cancer and normal samples, the RNA samples could be contaminated by microbes from the environment, such as microbes carried by equipment and personnel handling the samples and microbes present within reagents and kits. To identify contaminants, we adopted three different approaches: the identification of microbes that are highly abundant in certain stretches of sequencing dates, the identification of microbes that are highly abundant in certain plates used to store samples, and the identification of microbes whose read count does not increase following increases in the amount of total microbe reads.

We employed the first approach, because when visualizing microbes in a heatmap ordered by date sequenced, we observed microbes that exhibited a pattern of expression that is highly date dependent (Figure 2A,B). After the removal of microbes with visually conspicuous patterns, we discovered that most bacteria are of the phyla Actinobacteria, Firmicutes, and Proteobacteria (Figure 2C). For the contamination correction by plate approach, we found several microbes for which the expression is dramatically enriched in one or a few plates compared to other plates (Figure 2D). For the contamination correction by read counts, we found no microbes where increases in the total read count do not lead to an increase in the read count of a specific microbe (Figure 2E). The most significant correlation between total read count and the read counts of individual microbes was seen for uncultured bacterium. The read count of contaminants will not be expected to increase with total read count, because contaminants from the environment will most likely affect each sample by the same amount, while tissue-intrinsic microbes would be expected to be proportional in read count to the size of tissue or amount of genetic material sequenced.

### 2.4. Differential Abundance Analysis between Cohorts

After the extraction of bacterial abundance from each sample and the deletion of potential contaminants, we performed four differential abundance tests, comparing samples from each cohort to adjacent normal samples, using the Kruskal–Wallis test (*p* < 0.05) (Figure 3A). The cancer patient cohorts used were HBV+ drinkers, HBV+ nondrinkers, HBV− drinkers, and HBV− nondrinkers. This division of patients enabled an analysis of both the risk factors, that may affect the microbiome independently or together. After comparing the dysregulated microbes for each cohort against each other, we discovered that microbes dysregulated in HBV+/drinkers generally do not overlap with microbes dysregulated in the other three cohorts (Figure 3A, top row). Many microbes were similarly dysregulated between the other three cohorts (Figure 3A, bottom row). This result suggests that the double exposure to HBV and alcohol may result in a unique microbiome dysregulation landscape. When comparing the dysregulated microbes in each cohort in a four-way Venn diagram, we found that only two microbes, *Pantoea agglomerans* and *E. coli* 55989, are consistently dysregulated across all comparisons, which could suggest that these microbes’ dysregulation are central to LIHC (Figure 3B). Five different microbes are shared between the 3 cohorts other than HBV+/drinkers, while no other dysregulated microbe is shared between HBV+/drinkers and any other cohort besides the two shared across all comparisons. This result further highlights the difference in microbiome dysregulation landscape in HBV+/drinkers, compared to other cohorts. A closer examination of the dysregulations revealed that only three microbes were dysregulated in HBV+/drinkers, but significantly more microbes are dysregulated in the other cohorts (Figure 3C).

The majority of dysregulated microbes in the HBV+/nondrinker, HBV−/drinker, and HBV−/nondrinker cohorts are downregulated compared to adjacent normal samples (Figure 3C). In the presence of HBV only, *Pantoea agglomerans* is upregulated, but it is also upregulated in the control cohort without HBV or alcohol exposure. In the presence of alcohol only, three microbes are now upregulated: Enterobacter cloacae, Methylorubrum populi BJ001, and Rothia dentocariosa. Interestingly, *Pantoea agglomerans* is downregulated after exposure to alcohol.

For each patient cohort, a few microbes are uniquely dysregulated (Figure 3B). In the HBV+/drinker cohort, *Cutibacterium acnes* is uniquely upregulated. In the HBV+/nondrinker cohort, *Moraxella* sp. B and *Corynebacterium jeikeium* K411 are downregulated. In the HBV−/drinker cohort, *Methylorubrum populi* is upregulated and *Acidovorax ebreus* TPSY is downregulated. In the control cohort, *Staphylococcus epidermis* is downregulated.

### 2.5. Correlation between Microbial Abundance and Clinical Variables

We downloaded clinical variables data for each TCGA patient and correlated them to the abundance of microbes (Kruskal–Wallis, *p* < 0.05). We found, in all cohorts, an increased abundance of microbes correlates with poor prognosis, although a few exceptions to the trend do exist (Figure 4, Appendix A). Since most microbes are downregulated, the correlation of microbial abundance to clinical variables suggests that the dysregulation of the majority of bacterial species serves tumor suppressing effects. However, several microbes in the HBV+/nondrinker cohort correlate with the opposite trend, suggesting that, while the majority of dysregulations downregulate oncogenic microbes, a few tumor suppressive microbes are also downregulated after exposure to HBV. For the HBV−/drinker cohort, these potentially tumor-suppressing microbes include Acidovorax citrulli, which is downregulated and inversely associated with histologic grade; Sphingobium japonicum, which is downregulated and inversely associated with patient death; and Mycoplasma hyopneumoniae, which is downregulated and inversely associated with residual tumor presence (Figure 4B–D).

### 2.6. Association between Microbial Abundance and Cancer or Stem-Cell Associated Pathways and Genes

Associations between microbial abundance and pathway activities were quantified using gene set enrichment analysis (GSEA). For cancer pathways, we assessed the activity of pathways associated with critical regulators of cancer function, such as PTEN, RELA (NF-kB pathway), etc., using gene expression signatures. Signatures were downloaded from the Molecular Signatures Database (MSigDB) [15], where each signature contains the top 200 genes that are upregulated or downregulated after in vitro overexpression or the knockdown of a specific core cancer gene. After signature analysis, we found that a few microbes that are present in the HBV−/drinker cohort associated with the activities of multiple key cancer pathways (Figure 5A). *Caulobacter vibrioides* CB15 and *Herbaspirillum huttiense* subsp. putei IAM 15032 are associated with the upregulation of YAP1, TBK1, and the PRC2/EZH2 complex. *Anaerococcus prevotii* DSM 20548 is associated with upregulation BMI/PRC1 signaling. All three microbes are associated with increased *Pugh classification* grade, but not dysregulated compared to normal samples (Figure 4A). For the HBV+/nondrinker cohort, we found several microbes, namely *Staphylococcus epidermidis*, *Acinetobacter baumanni*, *Methylorubrum populi*, and *Acinetobacter calcoaceticus*, to be strongly associated with ATF2, AKT, and PIGF upregulation and P53 downregulation (Figure 5B). These microbes are also correlated with poorer prognosis, except for *Acinetobacter baumanni*, which is correlated with better prognosis. None of the microbes were dysregulated in cancer. Our analyses suggest that the presence of bacteria within the liver tumor microenvironment are often correlated with poorer prognosis, as well as the increased activity of cancer-associated pathways.

Overall, we observed that, within the HBV+/nondrinker cohort, the increased activities of the WNT pathway and the PTEN/AKT pathway are significantly associated with the upregulation of a large number of microbes, although most of the microbes also correlated with increased activities of stem cell pathways, specifically the YAP and embryonic stem cell differentiation pathways (Figure 5C). These correlations may indicate that WNT, PTEN, and AKT are master regulators of cancer and cancer stem cell-related processes in the HBV+ LIHC microbiome, which explains why a majority of microbes that were associated with both stem cell pathways and pathways associated with those three genes. However, within the HBV+/drinker cohort, we observed that, while many microbes are still associated with WNT, PTEN, and AKT signaling pathways, they no longer exhibit correlations with stem cell pathways (Figure 5D). Interestingly, a completely different set of microbes are associated with these cancer pathways in the HBV+/drinker cohort, compared to the HBV+/nondrinker cohort. These results may suggest that the alcohol in HBV+ LIHC patients could inhibit stemness through the regulation of the microbiome.

Finally, we correlated the dysregulation of microbes with the expression of canonical stem cell-promoting genes, including BMI1, SNAI1, SNAI2, TWIST1, CD44, VIM, and ALDH7A1. Consistent with our pathways analyses above, we find that the dysregulation of microbes in the HBV+/nondrinker cohort is associated with decreased expression of SNAI1 (Snail), SNAI2 (Slug), and VIM (Vimentin) (Figure 5E). Specifically, *Pantoea agglomerans* is upregulated in HBV+/nondrinkers and is correlated inversely with SNAI1 expression (high microbial abundance is correlated with low SNAI1 expression). Geobacillus sp. Y412MC61 abundance, which is downregulated in HBV+/nondrinkers, is directly correlated with SNAI2 and SNAI1 expression. Brachybacterium faecium DSM 4810 is downregulated in HBV+/nondrinkers, and its abundance is directly correlated with VIM expression. In the HBV+/drinkers cohort, we also observed correlation between the upregulation of certain microbes, namely *Ralstonia solanacearum* and *Escherichia coli* O111:H- str. 11128, and the decreased expression of SNAI2 (Figure 5F). However, another microbe, Streptococcus oralis, is upregulated in HBV+/drinkers, but also correlates with the increased expression of SNAI2. In conclusion, we found that microbiome dysregulation in HBV+/nondrinkers and HBV+/drinkers are mostly correlated with the decreased expression of stem cell genes, although the correlations are stronger and more consistent in patients exposed only to HBV. Microbiome dysregulation in the other two cohorts is not correlated with stem cell gene expression.

### 2.7. Correlation between Microbial Abundance and Immune System Modulation

The Cibersortx tool was used to determine the significant upregulation or downregulation of immune cell populations proximal to the tumor site (Figure 6A,B). In comparison to other cohorts, the HBV+/drinker cohort contains the greatest proportion of immune cell populations that correlated with microbe abundance, for both positive and negative correlations (Figure 6A,B). In the HBV+/drinker cohort, the existence of microbes in the liver was positively correlated with M1 and M2 macrophage populations, but negatively correlated with M0 macrophage populations. This indicates that the HBV+ microbiome may promote macrophage polarization. Increased levels of M1 and M2 macrophages may lead to inflammation that could drive carcinogenesis. Other immune cell populations that were positively associated with microbes in HBV+/drinkers included monocytes, neutrophils and activated NK cells. Similar to the HBV+/drinker cohort, microbes in the HBV−/drinker cohort were positively associated with both activated NK cells and M1 macrophage populations.

Because patients with HBV+ tumors had drastically different clinical outcomes from patients with drinker tumors, we used GSEA to analyze the enrichment of immune pathways in the HBV+/nondrinker and the HBV−/drinker cohorts, to further characterize differences between the etiologies. The HBV−/drinker cohort saw little immune pathway enrichment, with only *Caulobacter vibrioides* CB15 and *Herbaspirillum huttiense* subsp. putei IAM 15032 showing enrichment of immune signatures (Figure 6C). These microbes did not have any pathway enrichment in common. In the HBV+/nondrinker cohort, two microbes shown to be correlated with oncogenic and tumor suppressive signatures in Figure 5B, Acinetobacter calcoaceticus and Methylorubrum populi, had very similar profiles of immune signature enrichment (Figure 6D). Specifically, both microbes showed the upregulation of pathways relating to Treg activity, indicating that these microbes may account for the increase in Treg populations observed previously.

Using the software ReactomeFIViz, we plotted the top five most significant pathways that contain immune associated (IA) genes associated with microbes specific to each cohort (Figure 6E). The HBV+/drinker cohort was significantly associated with three immune pathways, including cytokine-cytokine receptor interaction, interleukin-10 signaling, and viral protein interaction with the cytokine and cytokine receptor. The HBV−/drinker cohort contained a greater proportion of cancer pathways (EGFR tyrosine kinase inhibitor resistance, Rap1 signaling pathway, cellular senescence and PD-L1 expression and PD-1 checkpoint pathway in cancer) than immune pathways (cytokine-cytokine receptor interactions). Similarly, the HBV−/nondrinker cohort also contained a greater number of significantly associated cancer pathways, including the ATF-2 transcription factor network and TNF signaling pathway, in comparison to immune pathways, where it was only associated with cytokine-cytokine receptor interaction. HBV+/nondrinker contained all cancer-related pathways.

ReactomeFIViz was also used to visualize functional interactions between inputted IA genes that were significantly associated with microbes within each cohort (Figure 6F). In the HBV+/drinker cohort, only microbes upregulated seemed to be associated with IA gene expression, as indicated by the red circular outline. Around half of this cohort’s significantly associated IA genes was positively associated with microbial abundance, while the other half was negatively associated. Interestingly, the microbes that were significantly correlated to the IA genes depicted in the other three cohorts (HBV−/drinker, HBV−/nondrinker and HBV+/nondrinker) were generally downregulated in their respective cancer cohort vs. normal comparisons, as seen by the blue circular outline. A mixture of positive and negative associations were still seen in these three cohorts.

## 3. Discussion

The microbiome has been significantly implicated in cancer development in recent years. Recent research has illustrated the microbiome as a powerful factor in influencing drug metabolism, inflammation, cancer progression, and cancer treatment outcomes [16]. However, the precise role of the microbiome in cancer is just beginning to be elucidated. Determining how microbes are involved in cancer is often a significant challenge, because they can either be harmful or beneficial [17]. A potential factor that determines whether a microbe is oncogenic or tumor suppressive is thought to be its effects on modulating the immune system, which could lead to chronic inflammation and an increased risk of cancer, but could also lead to immune activation, which is critical for eliminating cancerous cells when they arise.

In this study, we present provocative and unprecedented evidence that the liver microbiome is dysregulated in patients who are drinkers and who are infected with HBV, two major risk factors that are known to cause liver cancer development. Our study is among the first to demonstrate the existence of an intratumor microbiome in the liver. While multiple studies have investigated the gut microbiome-liver axis, no established evidence exists that proves that the liver harbors a microbiome in humans. However, it is likely that the liver could, because of its proximity to the gut. The liver receives drainage through the portal vein from the intestines, which accounts for 70% of blood influx into the liver [18]. While the portal vein should not be permeable to bacteria because of the gastrointestinal barrier, a number of diseases and conditions are associated with the disruption of this barrier, including inflammatory bowel disease, celiac disease, bowel obstruction, and gastrointestinal infection [19]. Breaching of this barrier has been associated with the entrance of microbes into the portal vein [19]. Other routes of origin of bacteria into the liver could also be conceived. A recent publication found that microbes migrate into the pancreas from the intestines through the pancreatic duct [20]. Through a similar mechanism, bacteria can theoretically enter the liver from the gut through the bile duct. There are also multiple reports of a biliary microbiome [21,22].

Exploring the association between the liver microbiome and two of the main etiological factors for LIHC, viral hepatitis B and heavy alcohol drinking, we discovered a significant microbiome dysregulation landscape, mediated by exposure to alcohol and HBV. Interestingly, we found that most microbes affected by alcohol and HBV exposure were downregulated compared to adjacent normal tissue, although there are a few exceptions. A relatively larger number of microbes was dysregulated after exposure to either HBV or alcohol than after exposure to both factors, and the microbes dysregulated by HBV or alcohol alone are also not similar to those dysregulated by the combined effects of HBV and alcohol. Therefore, HBV and alcohol combined may result in unique synergistic or antagonistic effects on the liver microbiome that have yet to be elucidated.

We also correlated the presence of different bacterial species with various clinical variables, including the *Pugh classification* grade, histologic grade, vital status, residual tumor, Ishak score for fibrosis, and pathologic stage. We discovered that, with few exceptions, most correlations with clinical variables were positive with poor prognosis. In other words, the increased abundance of most microbes was associated with poorer prognosis. Most microbes exhibiting strong correlation with poor prognosis are not dysregulated in LIHC or by either of the etiologies we examined. However, many microbes downregulated by alcohol or HBV also exhibited the same positive correlation with poor prognosis, raising the possibility that the dysregulation of these microbes is tumor suppressive. Since many microbes that are not dysregulated in alcohol and HBV-associated LIHC correlate with poor prognosis, we may infer that microbes that are already present in the liver environment could be contributing to tumor development. Indeed, many of these microbes’ abundance correlated with the increased activity of cancer-associated pathways, including the AKT, ERK, EZH2, and ATF2 pathways.

In addition to correlations with clinical variables and cancer pathways, we also examined the correlations between microbes and stem cell-related genes and microbes. In the cancer stem cell theory, a population of cells in the tumor exhibits stem cell features, and could self-renew, produce heterogeneous cancer cells, and orchestrate tumor growth, contributing to treatment resistance, metastasis, and relapse [23]. We found that, in HBV-exposed patients who do not drink, many microbes’ abundance positively correlated with increased activities of stem cell-related signatures. For the few microbes that are downregulated by HBV, their dysregulation associated strongly with decreased stem cell activity. Our results thus suggest that HBV-induced LIHC may downregulate cancer stem cell activity through downregulating microbe levels. However, the liver microbiome itself could contribute to increased cancer stem cells activity, thus contributing to tumor development and progression.

Based on our results, an important outstanding question is what determines if a tumor-promoting microbe can be downregulated by an etiological agent. The answer may be dependent on both the identity of the microbe and the etiological agent. For example, **Pantoea agglomerans**, which is known to cause infection in cancer patients [24], appears to be upregulated in LIHC samples from HBV+/nondrinkers and HBV−/nondrinkers. However, in samples from HBV−/drinkers, the same microbe is highly downregulated. Understanding how and why alcohol might induce the opposite trend of dysregulation for this microbe as that being induced by HBV may elucidate why certain microbes are downregulated in LIHC.

In order to elucidate the potential mechanism through which microbial abundance may be pro-tumor, we correlated microbial abundance to the expression of immune-associated genes in each of our cohorts. We discovered that cytokine signaling was strongly correlated with the abundance of microbes. For example, in the HBV+/drinker cohort, the upregulation of microbes correlated with the increased expression of cytokines, including CCL28, CCL26, CSF3, and SOCS3. However, several cytokines, like IL6 and IL10, were suppressed by microbe presence. Overall, a complex landscape of cytokine regulation might be conducted by the microbiome, as evidenced by the large number of microbial abundance correlations to cytokine expression.

In conclusion, we found that most microbes in the liver microbiome are tumor promoting in the context of HBV and alcohol exposure, but both factors could also downregulate microbes that are pro-tumor. However, only HBV, not alcohol, could downregulate microbes that may promote stem-cell function. These conclusions suggest different approaches to the diagnosis and management of HCC, depending on patient lifestyle and medical history. Recent findings have demonstrated the efficacy of blood serum diagnosis of cancer presence and risk using microbial signatures [25]. Additionally, future cancer treatments may aim to target the bioproducts of certain microbes to augment the immune system. Recent data may support the use of metabolites as markers for HCC, as HCC metabolism has been shown to have a unique signature. Specifically, glucose and acetate usage are significantly altered in HCC versus normal liver metabolism [26,27,28]. Our analysis has identified strains of **Escherichia coli** to be potentially important to HCC progression. Glucose and acetate concentration and metabolism play a key role in *E. coli* growth. When utilizing glucose to anaerobically grow, *E. coli* secretes acetate. The subsequent increase in concentration inhibits further *E. coli* growth without significantly changing *E. coli* metabolism [29]. This mechanism, called overflow metabolism, is hypothesized to be a common mechanism amongst fast-growing bacteria [30,31]. The growth-inhibiting role of acetate, coupled with anaerobic respiration, suggests a few possible mechanisms of liver dysbiosis. Changes to the gut microbiome may alter concentrations of acetate and glucose at the liver, thereby causing oncogenic dysbiosis. Alternatively, liver microenvironment changes in cancer may cause changes to bacteria levels. Dysbiotic bacteria may then alter concentration of metabolites and thereby affect other microbes and the tumor microenvironment at large. A further analysis of the metabolism of microbes identified in this study could possibly contribute to our understanding of HCC metabolism and therefore provide diagnostic and therapeutic opportunities. Further analysis of other gut disorders and their effect on the cancer microbiome may be important, as bacteria in the liver potentially originated from the gut. The effects of the microbiome on HCC could be attributed to manipulation by the gut microbiome, as is the case for breast cancer [32].

Several limitations of our study exist. First, the microbiome of healthy, normal livers may be hard to characterize because of the lack of sample availability. We were restricted to comparing microbes within the tumor to those in adjacent normal tissue, but it is conceivable that microbes in both regions might be the same once cancer develops. Second, direct alignment does not provide as high a resolution of microbe detection as 16s rRNA sequencing. Direct alignment is also restricted by already sequenced bacteria genomes, which are a fraction of the total microbiome. Finally, we would not be able to differentiate between live microbes in the liver and nucleic acid fragments from microbes in other parts of the body. Further in vitro and in vivo experimentations are needed to validate our findings and elucidate the mechanisms by which HBV and alcohol may mediate the microbiome composition.

## 4. Materials and Methods

### 4.1. Data Acquisition from TCGA

Raw whole-transcriptome RNA-sequencing data for tumor tissue were downloaded from the TCGA legacy archive [33] on 5 August 2018, for 373 LIHC samples and 50 adjacent normal samples. Level 3 normalized mRNA expression read counts for the above samples were downloaded from the GDC portal [34]. Clinical information for all patients were downloaded from the Broad GDAC Firehose [35].

### 4.2. Extraction of Microbial Reads and Calculation of Microbial Abundance

Using the Pathoscope 2.0 program [36], RNA-sequencing data were filtered for bacterial reads via direct alignment through a wrapper for Bowtie2. Bacterial sequences deposited at the NCBI nucleotide database [37] were used. Pathoscope generates two output measures quantifying the amount of bacterial species present in samples. One measure, best guess, quantifies the relative abundance of each species, expressed as a percentage. The other measure, best hit, signifies the absolute integer count of each species in the sequencing data.

### 4.3. Determination of Microbiome Diversity in Patient Samples

Using the Qiime2 framework [14], the best guess data output from Pathoscope were used to calculate alpha diversity and beta diversity using the *qiime diversity alpha* and *qimme diversity beta* modules respectively. A principle component analysis of the beta diversity results was done via the *qiime diversity pcoa* module and visualized using the *qiime emperor plot* module, the latter of which uses the EMPeror tool [38].

### 4.4. Differential Microbial Abundance between Cancer and Normal Patients

Differential abundance analysis was performed to compare microbe abundance (percentage abundance) in cancer tissues to microbe abundance in normal tissues of the same body site. Microbes that are present in less than ten percent of the patients in a cancer cohort were excluded. The Kruskal–Wallis analysis test was then applied to determine differential abundance (*p* < 0.05).

### 4.5. Correlation of Microbial Abundance to Survival and Clinical Variables

Survival analyses were performed while using the Kaplan–Meier model, with microbe expression being designated as a binary variable based on presence or absence of microbe in tumor samples. A univariate Cox regression analysis was used to identify candidates that were significantly associated with patient survival (*p* < 0.05). A clinical variable analysis was performed using the Kruskal–Wallis test, as described above.

### 4.6. Association between Microbial Abundance and IA Gene Expression

Using edgeR, a differential expression analysis was performed between mRNA expression of cancer and normal tissues to identify significantly dysregulated IA genes for each cancer (FDR < 0.05 and |log fold change| > 1). The Kruskal–Wallis test was used to correlate the abundance of significantly dysregulated microbes to significantly dysregulated IA genes (*p* < 0.05). Microbe abundance was modeled as a binary variable of presence and absence.

### 4.7. Identifying Connections between Significantly Dysregulated IA Genes

For each cohort, we filtered the significantly dysregulated IA genes identified from the previous analysis by *p*-value. The top 100 or so IA genes that had the most significant *p*-values when correlated to microbial abundance and were associated to microbes that were significantly dysregulated in the respective cancer cohort vs. normal were used. These IA genes were inputted into ReactomeFIViz which shows interactions between genes, to visualize the connections between the IA genes, in addition to finding pathways that contained these IA genes for each cohort.

### 4.8. Association between Microbial Abundance and Stem Cell and EMT-Associated Gene Expression

A panel of stem cell and EMT-associated genes were compiled by researching the literature. Using edgeR, a differential expression analysis was performed between mRNA expression of cancer and normal tissues, to identify significantly dysregulated stem cell and EMT-associated genes for each cohort (FDR < 0.05 and |log fold change| > 1). The Kruskal–Wallis test was used to correlate the abundance of significantly dysregulated microbes to significantly dysregulated stem cell and EMT-associated genes (*p* < 0.05). Microbe abundance was modeled as a binary variable of presence and absence.

### 4.9. Correlation of Microbial Abundance to Immune Infiltration

Estimated relative immune cell infiltration levels for 22 cell types were computed using the software CibersortX [39]. Microbe abundance was then correlated with immune cell infiltration levels for each microbe using the Kruskal–Wallis test (*p* < 0.05). Microbe abundance was modeled as a binary variable of presence and absence. The immune cell types examined include naïve B-cells, memory B-cells, plasma cells, CD8 T-cells, CD4 naïve T-cells, CD4 memory resting T-cells, CD4 memory activated T-cells, follicular helper T-cells, regulatory T-cells, gamma-delta T-cells, resting NK cells, activated NK cells, monocytes, M0-M2 macrophages, resting dendritic cells, activated dendritic cells, resting mast cells, activated mast cells, eosinophils, and neutrophils.

To determine the overall correlation of the microbiota of each cancer cohort with the infiltration levels of each immune cell type, the negative logged *p*-values for all correlations to each immune cell type were normalized as a fraction of the maximum negative logged *p*-value within each cancer cohort. For each immune cell population in each cancer cohort, the normalized log *p*-value were separately added based on whether they were correlatively negatively or positively to immune cell populations, in order to calculate the weighted sum.

### 4.10. Correlation of Microbial Abundance to Cancer and Immune-Associates Signatures

Signature enrichment corresponding to microbial abundance was measured using the gene set enrichment analysis (GSEA). Cancer and stem cell-associated signatures were chosen from the C6 set of signatures from the Molecular Signatures Database (MSigDB) [15]. Immune-associated signatures were chosen from the C7 set of signatures. Significantly enriched signatures were identified by a nominal enrichment score > 1 and a nominal *p*-value < 0.05. The direction of pathway enrichment was filtered to match the direction of clinical variable correlations per microbe.

### 4.11. Evaluation of Contamination Using Date of Sequencing

We applied a heuristic algorithm to extract sequencing dates where this overexpression occurs, which allowed us to determine potential contaminants’ relationship with the sequencing date. We visualized the microbial abundance of cancer patients in the form of a heat map and removed any microbe where stretches of dates with high microbial abundance exist, which we identified as contamination. In other words, contaminants are marked by marked non-uniform abundance across sequencing dates. For all the following analyses, we removed all microbes that were identified as contaminants.

### 4.12. Evaluation of Contamination Based on Plates

The abundance values of microbes were associated with plates on which the samples were stored prior to sequencing using the Kruskal–Wallis test (*p* < 0.05) and the visual examination of abundance differences between different plates using a boxplot.

### 4.13. Evaluation of Contamination Using Microbial Abundance Counts

The abundance of individual microbes in each patient is plotted against total microbe reads in the same patient, to determine if any microbe is likely a contaminant. Best hit results from Pathoscope are used for this analysis because absolute counts are required. In the resulting scatterplots, if a positive slope exists, it is likely that the microbe was biologically relevant and physically present in the sample, since the counts per microbe increased with the number of microbes sequenced. If the scatterplot has a slope of close to zero, and the counts of all the microbes are substantially above zero, it is likely that the microbe was a contaminant. This reasoning follows from the assumption that similar amounts of microbes will be present, regardless of how many microbes are present in the tissue sample if the microbe is an environmental contaminant. The Spearman correlation test and the correlation coefficient (*R*^2^) was used to calculate significance of a linear trendline and the slope of that trendline, respectively.

## 5. Conclusions

We conclude that variations to the intratumoral liver microbiome caused by alcohol and HBV significantly alter hepatocellular carcinoma progression. The nascent liver microbiome contains microbes that are pro-tumor when under the influence of alcohol and HBV, as evidenced by the poor prognosis and oncogenic gene expression associated with these microbes. However, the identity of the pro-tumor microbes is not the same for patient cohorts of different etiologies. HBV can also downregulate pro-tumor microbes, including those that are seemingly able to induce cancer cell stemness. Alcohol also downregulates pro-tumor microbes, but these microbes do not correlate with stemness. Additionally, the direction of dysregulation of certain microbes differs between drinkers and HBV+ patients, suggesting unique microbiota associated with each etiology. Finally, HBV and alcohol dysregulated microbes were also strongly associated with cytokine expression, indicating that these etiological factors may mediate cytokine levels through altering microbe levels.

## Figures and Tables

**Figure 1 cancers-12-01642-f001:**
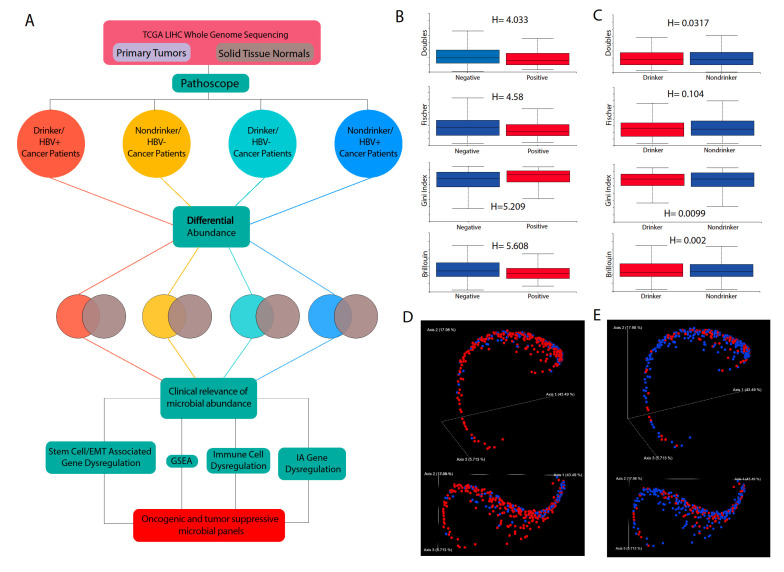
Overview of methods and patient data. (**A**) Schematic overview of the project methods. Comparisons of different alpha diversity measures (Doubles, Gini Index, Fischer, and Brillouin) between (**B**) hepatitis B HBV+ and HBV− patients and (**C**) drinker and nondrinker patients, with significant differences indicated by H-statistics > 1. Principle Component analysis plots of beta diversity per patient with (**D**) red representing HBV+ and blue representing HBV− and (**E**) red representing drinker and blue representing nondrinker patients.

**Figure 2 cancers-12-01642-f002:**
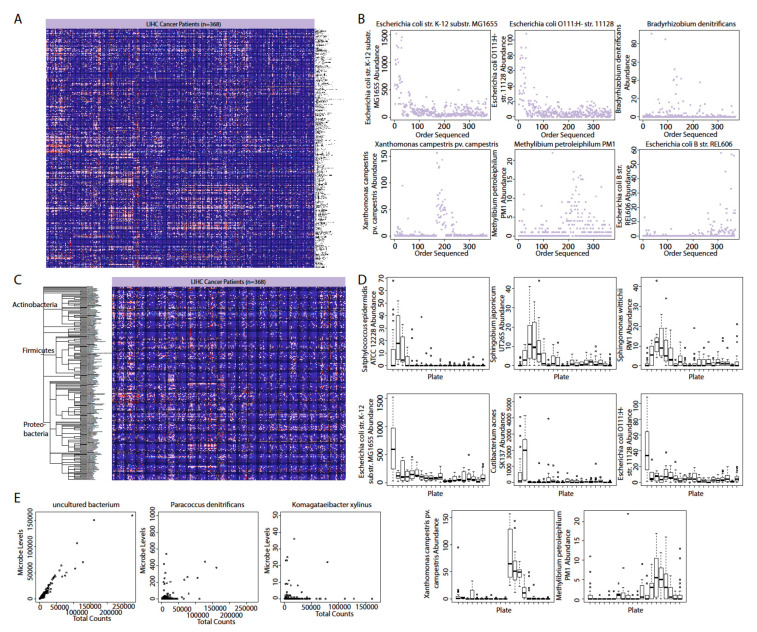
Overview of methods and patient data. (**A**) A heatmap of the original The Cancer Genome Atlas (TCGA) patient data prior to contamination correction. Contamination correction was first performed based on (**B**) sequencing date, with (**C**) potential contaminants removed from the analyses pool. Further correction was done based on (**D**) extreme changes in abundance between sequencing plates and (**E**) deviations in total read counts from a general average.

**Figure 3 cancers-12-01642-f003:**
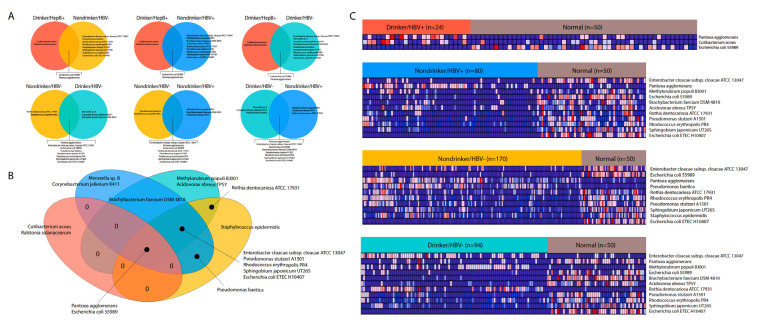
Differential abundance of microbes between cohorts. The Kruskal–Wallis test was used to compare the abundance of microbes and determine overlapping differential abundance between (**A**). two cohorts and (**B**). multiple cohorts. (**C**). Differentially abundant microbe abundance visualized in cancer, versus normal comparisons for each of the four cohorts.

**Figure 4 cancers-12-01642-f004:**
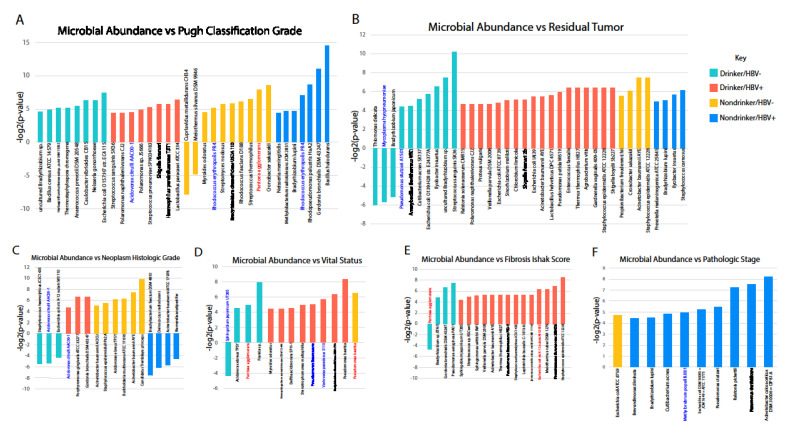
Correlation of microbial abundance to clinical variables. Microbial abundance was compared to (**A**) *Pugh Classification* grade, (**B**) residual tumor development, (**C**) initial tumor histologic grade, (**D**) vital status, (**E**) fibrosis Ishak score, and (**F**) pathologic stage. Bars with values greater than zero indicate a positive correlation with a worse clinical outcome per variable. Microbes labelled in red are upregulated in the cohort corresponding to the bar color while microbes labelled in blue are downregulated.

**Figure 5 cancers-12-01642-f005:**
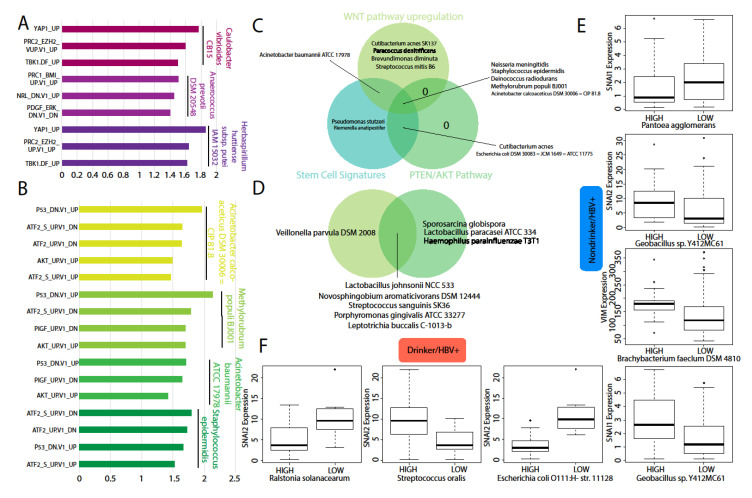
Correlation of microbial abundance to cancer-associated pathway expression. Downregulation of microbes in the (**A**) HBV−/drinker cohort and (**B**) the HBV+/nondrinker cohort are correlated to increased expression of cancer signatures that correlates with advanced clinical stages. (**C**) Clinically relevant microbial downregulation in the HBV+/nondrinker cohort is correlated to stem cell-associated signatures, which is not observed in (**D**) HBV+/drinker microbes. Co-regulation of stem cell-associated genes was observed in (**E**) the HBV+/nondrinker cohort and (**F**) the HBV+/drinker cohort.

**Figure 6 cancers-12-01642-f006:**
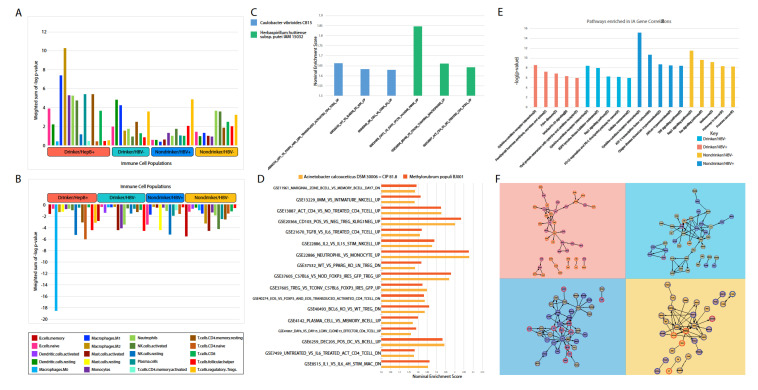
Correlation of microbial abundance to immune dysregulation. Immune cell population dysregulation in the (**A**) positive and (**B**) negative direction was analyzed for each cohort. Enrichment of immune signatures was further analyzed using gene set enrichment analysis (GSEA) on (**C**). HBV+/drinker and (**D**) HBV+/nondrinker data. (**E**) Top immune pathways dysregulated were identified by correlating immune associated genes to microbial abundance. (**F**) IA gene interaction diagram of genes correlated with microbe abundance in each cohort. Red or blue outlines of the circle indicate whether the abundance of the microbe associated with the IA gene is upregulated or downregulated in its respective cancer cohort vs. normal comparison, respectively. Orange or purple fill demonstrate that microbial abundance is positively or negatively correlated with IA gene expression, respectively.

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
