# Peer review of "The Liver Microbiome Is Implicated in Cancer Prognosis and Modulated by Alcohol and Hepatitis B"

_cancers, 2020, doi:10.3390/cancers12061642_

Round 1
Reviewer 1 Report
The Ms by Chakladar et al. showed a computational analysis of TCGA data on the relevance of microbiome in the pathogenesis of hepatocellular carcinoma (HCC), in particular in HCC cases with HBV and alcohol etiologies. The authors screened the presence of microbes (via nucleic acid identification) in the liver, but moreover whether that they could be associated with clinical parameters of the disease. I agree with the statement of the authors that this study is provocative, and I have several issues needed to be addressed by the authors.
- The presence and the role of microbiome in disease progression are complex since they are very heterogeneous. In fact, looking at the data (e.g. Figure 3), I observed high variations of microbes not only within cohort, but also between cancerous and no-cancerous tissues and among different etiologies. In the Ms, the authors performed an extensive bioinformatics study correlating the microbial panels with oncogenic pathway, stem cell, immune dysregulation, etc. I understand the due to data limitation, they did not check healthy donor liver, but did the authors cross-checked these molecular targets with the dysregulation of microbiome in other tumors with the known presence of microbes (intestine cancer, colorectal cancer, etc.)?
- It is known in pathological condition, the pathological bacterial translocation (BT) stimulates an augmented pro-inflammatory cytokine response and release of ROS and NOx (Wiest et al., J Hepatol 2014), resulting in the progression of the disease due to BT. Nevertheless, the authors showed down-regulation for several microbes and stated that they were cancer-suppressive, what would be the explanation for this data?
- Further, the possible-association between the microbial abundance and clinical variables is not clear (Figure 3). What microbe is associated with poor prognosis? How long is the survival value (month, Kaplan Meier analysis)? What are the histological score (Edmonson Steiner?), fibrosis score (F1-F4?), Child-Pugh indicated to be relevant with the microbial abundance?
- It would be good if the authors could provide the significance of this study in the management of the disease.
Author Response
Reviewer 1
The Ms by Chakladar et al. showed a computational analysis of TCGA data on the relevance of microbiome in the pathogenesis of hepatocellular carcinoma (HCC), in particular in HCC cases with HBV and alcohol etiologies. The authors screened the presence of microbes (via nucleic acid identification) in the liver, but moreover whether that they could be associated with clinical parameters of the disease. I agree with the statement of the authors that this study is provocative, and I have several issues needed to be addressed by the authors.
- The presence and the role of microbiome in disease progression are complex since they are very heterogeneous. In fact, looking at the data (e.g. Figure 3), I observed high variations of microbes not only within cohort, but also between cancerous and no-cancerous tissues and among different etiologies. In the Ms, the authors performed an extensive bioinformatics study correlating the microbial panels with oncogenic pathway, stem cell, immune dysregulation, etc. I understand the due to data limitation, they did not check healthy donor liver, but did the authors cross-checked these molecular targets with the dysregulation of microbiome in other tumors with the known presence of microbes (intestine cancer, colorectal cancer, etc.)?
- It is known in pathological condition, the pathological bacterial translocation (BT) stimulates an augmented pro-inflammatory cytokine response and release of ROS and NOx (Wiest et al., J Hepatol 2014), resulting in the progression of the disease due to BT. Nevertheless, the authors showed down-regulation for several microbes and stated that they were cancer-suppressive, what would be the explanation for this data?
- Further, the possible-association between the microbial abundance and clinical variables is not clear (Figure 3). What microbe is associated with poor prognosis? How long is the survival value (month, Kaplan Meier analysis)? What are the histological score (Edmonson Steiner?), fibrosis score (F1-F4?), Child-Pugh indicated to be relevant with the microbial abundance?
- It would be good if the authors could provide the significance of this study in the management of the disease.
- Thank you for your insightful comments. We do realize that the microbiome Is highly complex and heterogenous, so we have examined the potential role of the microbiome in ten other cancers beside liver cancer and compared them to each other. These ten cancers include cancers in the bladder, lung, head and neck, breasts, prostate, pancreas, colon, rectum, and cervix. These sites were chosen because they were traditionally known to contain microbes or recently reported to contain microbes in landmark papers. The same analysis procedure as employed in this manuscript was employed for these 10 cancers, although we only compared cancer samples to adjacent normal samples. We discovered that in all the tissue sites, cancer samples always contain more microbes than normal samples, and there is significant differential abundance for many microbes. Therefore, we believe that there is evidence that microbe abundance levels at cancer sites are a result of cancer altering the microbiome, and comparing cancer tissue to adjacent normal tissue would give us the best insight into what kind of alteration has occurred. Even though we do not have healthy liver samples, we think adjacent normal samples are adequate for us to ascertain the role of cancer in dsyregulating the microbiome. Comparing the abundance of bacteria between bacteria in liver cancer and bacteria in other cancers, we discovered a significant degree of commonality on every taxonomical level, suggesting that there is some consistency in which microbes are present within tumors. This result affords us with some confidence to the validity of our results. However, we did not present the data of inter-cancer comparison in this manuscript because we plan to publish the data for other cancers in a separate study. We did publish the results of our analysis on the lung cancer microbiome’s association with age and gender recently (Wong et. al, Cancers, 2020), where we observed some of the same species and genera of bacteria dysregulated in liver to be dysregulated in lung, such as E. coli, R. dentocariosa, and Pseudomonas.
- These microbes that were downregulated were not tumor suppressing but tumor promoting, which is consistent with the paper you cited. However, because these tumor-promoting microbes were downregulated, the effect of dysregulation is essentially tumor suppressing. It is true that many bacteria can promote tumor progression through promoting inflammation or generating ROS, and we agree that most bacteria we found correlate with unfavorable prognosis through clinical variable correlations. However, we found that some of these bacteria were downregulated, so we wanted to highlight this interesting finding. Most bacteria correlated with poor prognosis were actually upregulated in liver cancer, but we highlighted the downregulated ones, which were also some of the most highly differentially abundant species, to challenge the idea that tumor promoting bacteria are always upregulated in cancer. The mechanism by which some bacteria are downregulated while others are upregulated should be further elucidated. We have specified this rationale in the Discussion.
- Microbes correlated with poor prognosis are those that have positive values on the bar plots in Figure 4. Positive bars indicate a correlation with more advanced clinical stages and higher patient mortality. We have presented some of the most significant correlations in boxplot format in Supplementary Figure 1. A trend on the boxplot where high microbe abundance is correlated with poorer prognosis would correspond to positive bars in Figure 4, whereas if low microbe abundance is correlated with poorer prognosis, the bars will point in the negative direction.
- That is an excellent suggestion. We have added a discussion of disease management in section 3 (Lines 384-389)

Reviewer 2 Report
Chakladar et al. tried to dissect the liver microbiome in HCC using RNA-sequencing data from The Cancer Genome Atlas (TCGA).They classified HCC into four categories with HBV and alcohol status and found that HBV and alcohol cooperate to promote HCC development with some oncogenic microbes, although the other oncogenic microbes could be downregulated.
Liver microbiome, not gut microbiome, is still controversial for its oncogenic property, however, this is an interesting study to try to clarify the association between them by analyzing existing data sets. However, I have some concerns with this study, and I cannot recommend it for publication in the present form.
Major comments.
1) The authors classified HCC into four categories with HBV and alcohol status, however its rationale is not clear and need to be mentioned. Fig. 1B and C; What are these four kinds of bar chart? Need to explain. How about other etiologies such as HCV?
2) Fig 3C. The authors compared the microbes between the data derived from tumor and non-tumor samples, however the comparison between tumor and non-tumor corresponding to tumor (obtained from same patient) is missing. How was the abundancy of microbes normalized between different data sets? How was the result between tumor and non-tumor samples derived from same patient?
Minor comments.
1) Some references are not suitable for the context.
For example, Line 45; Ref 1. is about microRNA-155 and HCC and is not appropriate for the sentence "Hepatocellular carcinoma (HCC) is the most common form of liver cancer and the second 45 deadliest cancer, affecting over 500,000 people worldwide every year."
Line 198; Ref 3. is about epidemiology of HCC and not mentioned about MSigDB.
The authors need to be careful to choose appropriate manuscripts when preparing the references.
2) Discussion; According to this manuscript, some bacterial species may be oncogenic, and some may be not. The authors need to discuss the obtained results about microbiome specifically based on the previously published manuscripts, that will help the readers to understand the meanings of this study.
Author Response
Reviewer 2
Chakladar et al. tried to dissect the liver microbiome in HCC using RNA-sequencing data from The Cancer Genome Atlas (TCGA).They classified HCC into four categories with HBV and alcohol status and found that HBV and alcohol cooperate to promote HCC development with some oncogenic microbes, although the other oncogenic microbes could be downregulated.
Liver microbiome, not gut microbiome, is still controversial for its oncogenic property, however, this is an interesting study to try to clarify the association between them by analyzing existing data sets. However, I have some concerns with this study, and I cannot recommend it for publication in the present form.
Major comments.
1) The authors classified HCC into four categories with HBV and alcohol status, however its rationale is not clear and need to be mentioned. Fig. 1B and C; What are these four kinds of bar chart? Need to explain. How about other etiologies such as HCV?
2) Fig 3C. The authors compared the microbes between the data derived from tumor and non-tumor samples, however the comparison between tumor and non-tumor corresponding to tumor (obtained from same patient) is missing. How was the abundancy of microbes normalized between different data sets? How was the result between tumor and non-tumor samples derived from same patient?
Minor comments.
1) Some references are not suitable for the context.
For example, Line 45; Ref 1. is about microRNA-155 and HCC and is not appropriate for the sentence "Hepatocellular carcinoma (HCC) is the most common form of liver cancer and the second 45 deadliest cancer, affecting over 500,000 people worldwide every year."
Line 198; Ref 3. is about epidemiology of HCC and not mentioned about MSigDB.
The authors need to be careful to choose appropriate manuscripts when preparing the references.
2) Discussion; According to this manuscript, some bacterial species may be oncogenic, and some may be not. The authors need to discuss the obtained results about microbiome specifically based on the previously published manuscripts, that will help the readers to understand the meanings of this study.
Major comments
- The 4 cohorts are: HBV+ drinkers, HBV+ nondrinkers, HBV- drinkers, and HBV- nondrinkers. We have added a more detailed description of the four categories to make the division clearer, specifying why the 4 cohorts were chosen. (lines 143-144). For Fig 1B and C, the labels from the Y-axis were missing. We added them in to indicates that these box plots are comparing different measures of alpha diversity. We have also updated the figure caption with this information (Line 107). We chose to study the effects of alcohol and HBV because these risk factors are some of the most common for HCC patients in general and for TCGA HCC patient specifically. Therefore, the large number of patients with either HBV+ or drinker status enabled us to have more powerful statistical analyses. It is a very good suggestion to examine HCV. As it is known that both HCV and HBV cause HCC via gradual cirrhosis of the liver, we would like to eventually understand if the HCV-associated microbiome is similar to the HBV-associated microbiome in terms of the way that it is key to HCC pathogenesis. However, we chose not to include HCV in this study primarily because it would have been difficult to compare between 3 etiologies for a study this size, as it would add at least 2 more cohorts for analysis. In a future study, we plan to compare the HCV+ microbiome to HBV+ microbiome, possibly by supplementing TCGA data with more HCV+ and HBV+ HCC samples from other studies. The effects of the fourth most common risk factor, non-alcoholic fatty liver disease, are difficult to analyze using TCGA data alone, as there are few patients with recorded non-alcoholic fatty liver disease. For future projects, if we were to implicate this etiology, it would require us to procure sequencing data from a different study.
- For this study, the normal samples used were all adjacent normal from TCGA data. These tissues were dissected beyond the margins of the cancer tissue during surgery. Therefore, comparisons between cancer and normal are essentially a comparison of tumor vs non-tumor tissue from the same patient at the same site. TCGA has already normalized sequencing data, so the microbial abundance was not normalized again.
Minor Comments
- We have removed the first reference, adjusted the second reference, and have ensured that the other references are accurate.
- We appreciate this comment and have added a discussion about this topic in Section 1 (lines 71-75)

Reviewer 3 Report
In this work, Chakladar and coll. attempt to investigate the effects of virus B-linked hepatocarcinoma (HBV) and alcohol on the liver microbiome. To this aim they analyzed microbial abundance after correction of potential microbe contaminants.
Although the work is of interest, some concerns have to be taken into account. As the authors stated, the presence of bacteria in the liver is always secondary to upcoming gut pathology. Thus, changes in the tumor microbiome could mirror the gut disease and not be associated with the tumor. The authors have to consider a recent review on the topic by Vergara et al. (Vergara D, et al .J Oncol. 2019 Oct 20;2019:1253727. doi: 10.1155/2019/1253727).
However, the authors have to consider the metabolic effect of some microbe metabolites such as acetate, butyrate, etc. on cancer metabolism. This aspect could be of interest and could be inserted in the discussion based on the recent review reporting the metabolic changes of HCC (De Matteis S, et al. Oxid Med Cell Longev. 2018 Nov 4;2018:7512159. doi: 10.1155/2018/7512159. eCollection 2018.)
Minor considerations:
The authors have to carefully check for any typing error (i.e. Fig. 1A in paragraph 2.4 probably can be Fig. 3).
Author Response
Reviewer 3
In this work, Chakladar and coll. attempt to investigate the effects of virus B-linked hepatocarcinoma (HBV) and alcohol on the liver microbiome. To this aim they analyzed microbial abundance after correction of potential microbe contaminants.
Although the work is of interest, some concerns have to be taken into account. As the authors stated, the presence of bacteria in the liver is always secondary to upcoming gut pathology. Thus, changes in the tumor microbiome could mirror the gut disease and not be associated with the tumor. The authors have to consider a recent review on the topic by Vergara et al. (Vergara D, et al .J Oncol. 2019 Oct 20;2019:1253727. doi: 10.1155/2019/1253727).
However, the authors have to consider the metabolic effect of some microbe metabolites such as acetate, butyrate, etc. on cancer metabolism. This aspect could be of interest and could be inserted in the discussion based on the recent review reporting the metabolic changes of HCC (De Matteis S, et al. Oxid Med Cell Longev. 2018 Nov 4;2018:7512159. doi: 10.1155/2018/7512159. eCollection 2018.)
Minor considerations:
The authors have to carefully check for any typing error (i.e. Fig. 1A in paragraph 2.4 probably can be Fig. 3).
1-2) These are excellent suggestions that have enabled us to further contextualize our current study and potential future studies. After reading the reviews by Vergara et al and De Matteis et al, we added to the Discussion section information about the gut-liver axis and hypothesize how it may relate to the HCC metabolome. It is known that acetate and glucose concentrations were altered significantly in HCC versus the normal liver microenvironment. Further reading into these two metabolites revealed an important mechanism used by fast-growing bacteria: overflow metabolism. This mechanism is a method of growth control, where anaerobic respiration of glucose produces acetate that prevents further growth of bacteria utilizing overflow metabolism. This discussion is very important to our paper, as we have identified strains of E. coli, a bacteria that uses overflow metabolism, to be important. Further investigation of this mechanism may be important to understanding the HCC microbiome, as we have discussed in the manuscript (lines 393-410).
3) We have fixed the indicated typing error and have corrected any others found in the manuscript.

Round 2
Reviewer 1 Report
Major comment: I have no any other major issues on the Ms.
Minor comment: Please increase the figure font size (especially on Figure 2 and Figure 5 box plot).
Author Response
We greatly appreciate your evaluation of our manuscript. We have significantly increased the font size of the indicated figures.
Reviewer 2 Report
The revised manuscript is much improved; the authors have addressed my issues. I recommend its acceptance in present form.
Author Response
Thank you very much for your comments.